# Pet Ownership and Mental and Physical Health in Older White and Black Males and Females

**DOI:** 10.3390/ijerph19095655

**Published:** 2022-05-06

**Authors:** Amy E. Albright, Ruifeng Cui, Rebecca S. Allen

**Affiliations:** 1Department of Veterans Affairs, VA Maine Health Care System, Augusta, ME 04240, USA; 2Department of Psychology, The University of Alabama, Tuscaloosa, AL 35401, USA; rsallen@ua.edu; 3Alabama Research Institute on Aging, The University of Alabama, Tuscaloosa, AL 35487, USA; 4Department of Veterans Affairs, VISN 4 Mental Illness Research, Education and Clinical Center, VA Pittsburgh Health Care System, Pittsburgh, PA 15240, USA; ruifeng.cui@va.gov; 5Department of Psychiatry, School of Medicine, University of Pittsburgh, Pittsburgh, PA 15213, USA

**Keywords:** pet, older adult, depression, health, activity

## Abstract

Pet ownership literature remains mixed regarding associations with mental and physical health outcomes among older adults. The present study investigates the relationship between pet ownership and depression, health, and physical activity in an older adult sample balanced by sex (male/female), race (White/Black), and urban/rural status. Participants were adults aged 65+ recruited between 1999 and 2001 in the University of Alabama at Birmingham Study of Aging. Participants completed the Geriatric Depression Scale, a single-item self-reported health measure, and a physical activity questionnaire. Dog owners reported better subjective health and were more likely to walk for exercise as compared to non-pet owners. Cat owners did not differ from non-pet owners in terms of self-reported health or walking. White participants were more likely than Black participants to report ownership of a pet. No relationships were found between pet ownership and symptoms of depression. Findings were not influenced by sex, race, or geographical location. Dog ownership may be associated with positive physical health behaviors and subjective health perceptions. Additional research focused on mechanisms and cognitive impact is needed. Although there may be physical health benefits of dog ownership, adopting a pet should not be viewed as a simplistic solution to alleviating depression in older adults.

## 1. Introduction

Pet ownership among adults is common, with 38–48% of adults owning dog(s) and 25–38% of adults owning cat(s) in the United States [1]. Additionally, rates of pet adoption have increased significantly during the current coronavirus pandemic [2]. However, the literature on the positive benefits of pet ownership on the mental and physical well-being of older adults is mixed. Pet ownership conveys unique struggles and responsibilities which may be exacerbated by global pandemics [3]; consequently, additional research is warranted to determine the potential health benefits of pet ownership to inform decision making among individuals seeking to adopt a pet to improve their mental or physical health.

Depression is associated with a host of negative mental and physical health outcomes among older adults [4,5,6], with a prevalence rate of up to 5% [4]. Research on pet ownership among older adults suggests that owning a pet may help alleviate loneliness and improve feelings of depression [7]. Although some studies have found beneficial associations between pet ownership and depression among older adults, this literature remains mixed [8,9]. Of note, the distinction between animals kept as pets and those trained in animal-assistance must be made. While animal-assisted therapy may have significant positive impacts on mental health in older adults, (i.e., [10]), the literature is again mixed on whether typical pets without specialized training may provide a similar effect. 

In addition to potential mental health benefits, pet ownership may be associated with various physical health benefits. Research on the relationship between pet ownership and physical health has mostly been conducted with younger adults and has found positive relationships between dog and cat ownership and measures of physical health [11]. The limited literature on pet ownership among older adults has focused on the benefits of dog ownership, as older adult dog owners have been shown to have significantly higher levels of physical activity than older adults who did not own a dog or did not report walking their dog [12]. They also tend to be healthier than cat owners, even when other factors, such as behavioral and physical health, are controlled [13,14].

Older adult pet owners may differ from older non-pet owners on several demographic characteristics such as sex, race, and geographical location. Further, these demographic characteristics may moderate and thus help explain the mixed findings in the relation between pet ownership and mental and physical health. To date, only one study investigated whether sex, race, and geographical location moderated the relation between pet ownership and health; however, the sample was predominantly young to middle aged [15]. This cross-sectional study found that Blacks and individuals living in urban settings were less likely to own dogs or cats as compared to Whites and those living in rural settings. Owning dogs or cats was not associated with overall health but was associated with more depression symptomatology than non-pet owners and these findings were not moderated by sex, race, or geographical location.

Using secondary data from the eight-year-long University of Alabama Birmingham (UAB) Study of Aging taken from the 84- and 90-month assessment time points, the aim of the current study was to assess the relationship between symptoms of depression, self-reported health, and pet ownership in older adults. The study seeks to extend the limited literature on the type of pets owned and depression and physical health outcomes among older adults. The study is unique in that it is the first to investigate the potential benefits of pet ownership in a longitudinal sample of older adults balanced by sex (male vs. female), race (White vs. Black), and geographical location (urban vs. rural residence) which allows for the investigation of whether the potential benefits of pet ownership in later life differ by these individual characteristics. This is a particularly important area of investigation given that Black and rural-dwelling individuals are frequently excluded from research, and thus may be underrepresented in the literature [16]. Older adults who were pet owners were predicted to have lower symptoms of depression as well as higher self-reported health and greater levels of physical activity as compared to older adults who do not own any pets.

## 2. Materials and Methods

The study is a secondary analysis of data from one thousand participants from Alabama who were recruited between 1999 and 2001 for the UAB Study of Aging and then subsequently followed for eight years. All participants were community-dwelling and at least 65 years of age at the time of initial recruitment. The study was designed to provide a sample balanced by sex, race, and urban/rural residence. The study was conducted according to the guidelines of the Declaration of Helsinki and approved by the Institutional Review Board of UAB (protocol code IRB-160107005). Secondary data analyses were approved by the Institutional Review Board of the University of Alabama (protocol code IRB-4123). Informed consent was obtained from all subjects involved in the study. Participants were 1000 Medicare beneficiaries randomly recruited by letter from two urban and three rural counties located within central Alabama. All participants were community-dwelling, as residents of skilled nursing facilities were excluded from this study. In addition, individuals who were not able to independently schedule study appointments were excluded. Following obtainment of informed consent, an initial interview that lasted for approximately two hours was conducted by trained personnel within the participant’s home to collect initial baseline data. Following the initial interview, follow-up telephone interviews were conducted every six months for eight years [17]. While data collection for this study began approximately twenty years ago, issues of depression and loneliness continue to be significant issues faced by older adults [6], and this topic remains highly relevant in the current era of COVID-19. As noted, physical and mental health concerns may be exacerbated by social isolation due to the current pandemic [18].

### 2.1. Measures

Pet Ownership. Data regarding pet ownership were taken from the 84-month assessment, when participants were asked if they had any pets, and if so, what type of pet (i.e., “dog”, “cat”, “bird”, or “other”). If the pet was a dog, the participant was then asked if they walked the dog, and if so, how many times and the average duration of the walks over the past two weeks. Current analyses excluded the “other” category as only 3 participants endorsed this response and no participants endorsed bird ownership.

Depression. The 15-item Geriatric Depression Scale GDS-15 [19] was used to collect information regarding depression at the 84-month assessment timepoint. The GDS-15 is a measure of depressive symptomatology, designed and normed for use with older adults. Items are phrased as yes/no questions and the final score was computed as a sum of all items with higher scores indicating greater depressive symptomatology. Of note, the GDS-15 is a screening measure, and, while it measures depressive symptomatology, clinically significant scores (i.e., greater than 5) are not equivalent to a diagnosis of major depressive disorder according to DSM-5 criteria. 

Self-Reported Health and Physical Activity. These constructs were assessed using questions created by the original authors of the UAB Study of Aging [17]. Self-reported health data from the 84- and 90-month assessment time point was used in the current study. Participants were asked to rate their general health via a single question on a 5-point Likert scale, with options ranging from 1 = “Excellent” to 5 = “Poor.” Lower scores represent higher levels of self-reported health. Information about physical activity was taken from the 84-month assessment time point. Participants were also asked one question regarding if they had walked for exercise within the past year. If they indicated yes, they were then questioned regarding the average amount of time they walked on each occasion, the number of months in a year they walked, and how many times they walked during the past two weeks. While walking for exercise was assessed as a separate variable from walking with a pet dog, it is likely that the walking for exercise variable also captures participants who exercised with their pets. 

### 2.2. Analyses

Descriptive statistics were used to characterize the sample of dog owners, cat owners, and non-pet owners. Chi-squares and ANOVAs were used to analyze differences in participant demographic and health characteristics by pet ownership status. Follow-up moderation interaction analyses were conducted for all significant variables to assess whether findings were moderated by sex, race, and geographical location. Data were analyzed using IBM SPSS Version 27 [20].

## 3. Results

At 84 months, data on pet ownership were available for 535 older adults. Participants were on average 73.3 years old (SD = 5.6), with slightly more females (N = 292, 54.6%), Whites (N = 277, 51.8%), and older adults living in rural areas (N = 281, 52.5%). See Table 1 for additional descriptive characteristics.

Most participants did not own pets (N = 360, 67.3%), with 21.7% owning dogs (N = 116) and 11.2% owning cats (N = 60). Compared to Black participants, White participants were more likely to own cats (14.4% of White participants owned a cat vs. 7.6% of Black participants owned a cat; *X*^2^(1, N = 419) = 7.7, *p* = 0.01) and dogs (24.2% of White participants vs. 19.0% of Black participants; *X*^2^(1, N = 475) = 3.8, *p* = 0.05). Dog and cat owners did not significantly differ in terms of geographic location. Older adults living in urban areas were not more likely to own a dog or a cat as compared to older adults in rural areas and vice versa. Similarly, sex did not influence type of pet owned, as males and females did not significantly differ from one another on their likelihood of owning a cat or a dog. Participants with dogs reported between 0 and 14 walks with their dogs in the two weeks prior to data collection (M = 0.71, SD = 2.84), and these walks were reported to last between 0 and 360 min (M = 4.79, SD = 31.74). However, of the 116 dog owners, only 9 participants indicated that they had walked with their dog at any point within the past two weeks.

At 84 months, data indicated a sex difference in self-rated health, with females reporting better subjective health (ANOVA F(1, 531) = 9.53, η_p_^2^ = 0.018, *p* < 0.01). There was a similar effect of sex on depressive symptomatology, with females reporting higher GDS-15 scores (ANOVA F(1, 507) = 4.67, η_p_^2^ = 0.009, *p* = 0.03). Depression symptomatology also differed between Whites vs. Blacks, as Black participants reported higher GDS-15 scores (ANOVA F(1, 507) = 23.14, η_p_^2^ = 0.044, *p* < 0.01). Similarly, individuals living in rural locations also had significantly higher GDS-15 scores than their urban counterparts (ANOVA F(1, 507) = 17.28, η_p_^2^ = 0.012, *p* = 0.01). See Table 2 for additional descriptive statistics on health ratings by sex, race, and geographical location.

The sample had overall low rate of depression symptomology (M = 1.7, SD = 1.7), and pet ownership status (i.e., no pets, owns dogs, owns cats) at 84 months was not associated with depression symptoms at 84 months (*p* > 0.05 for all). 

A small proportion of the sample (N = 67, 12.6%), reported walking for exercise in the past year at the 84-month assessment. These participants typically exercised 10.9 (SD = 2.6) months of the year, with walks lasting between 5 and 60 minutes (M = 27.3, SD = 10.9). In the prior two weeks before the 84-month assessment, participants reported walking on average 7.6 times (SD = 4.6). Dog owners were more likely to walk for exercise in the past year than non-pet owners (*X*^2^(1, N = 473) = 4.30, *p* = 0.04). This effect was not moderated by sex, race, or geographical location (*p* > 0.05 for all interactions). Among individuals who walked for exercise, however, dog owners did not differ from non-pet owners in the frequency or duration of walking for exercise (*p* > 0.05 for all). Cat owners did not differ from non-pet owners in whether they had walked for exercise during the last year (*p* = 0.06). 

Individuals who were dog owners at 84 months reported better health (M = 3.20, SD = 0.58) than non-pet owners (M = 3.34, SD = 0.64; ANOVA F(1, 471) = 4.47, η_p_^2^ = 0.009, *p* = 0.04). However, dog ownership did not predict better self-reported health six months later at the 90-month follow-up, (ANOVA F(1, 452) = 3.5331, η_p_^2^ = 0.009, *p* = 0.06). Repeated measures ANOVA predicting self-reported health across 84 months and 90 months found a significant main effect for dog ownership, i.e., owned a dog vs. did not own a dog, (F(1, 451) = 6.37, η_p_^2^ = 0.014, *p* = 0.01) and time, i.e., change in self-reported health from 84 months to 90 months (F(1, 451) = 6.04, η_p_^2^ = 0.013, *p* = 0.01). Sex, race, and geographical location did not moderate the effect of dog ownership on self-reported health (*p* > 0.05 for all interactions). Cat owners did not differ from non-pet owners in ratings of self-reported health at 84 or 90 months (*p* > 0.05 for all).

## 4. Discussion

The present study investigated the relationship between pet ownership and mental and physical health in an older adult sample balanced for sex, race, and geographical location. Older adult females reported on average better overall health but worse depression symptoms as compared to older males, which is consistent with prior research regarding self-reported health [21] and depression [22]. Black older adults and older adults living in rural locations reported on average worse depression symptoms as compared to Whites and individuals living in urban locations, respectively. This is consistent with prior research, which suggests that older adults living in rural locations are more likely to report symptoms of depression [23]. The literature is more mixed regarding depression in older Black adults, as Whites are more likely to be diagnosed with depression; however, it is likely that formal diagnoses are influenced by social determinants of health, and depression may simply be less recognized in minority populations [24]. Given that the current study used the GDS-15 to assess symptoms of depression rather than a formal diagnosis of a mood disorder, symptomatology may be better captured by this measure and may explain higher rates of self-reported symptoms in Black participants. 

White older adults were more likely to own dogs and cats than Black older adults. While it is possible that this is potentially explained by discrepancies in income, as pets may be a significant expense, data regarding self-reported income is unfortunately not available at the 84-month timepoint and remains an area for potential future study. Dog owners were more likely to have walked for exercise in the prior 12 months and reported better overall health as compared to non-pet owners. It is important to note that although dog owners tended to walk more for exercise, only a small minority of them, 7.8%, reported walking their dogs over a two-week period. This was a surprising finding, given that increased exercise in the company of a pet is often a reason for adoption. However, research has demonstrated that access to yards or public outdoor spaces allows many dog owners to provide adequate exercise for their pet without being required to take them for walks [25]. It is likely that this explains, at least in part, the low number of participants who reported walking with their dogs, as these spaces tend to be fairly common and accessible in central Alabama. In contrast, cat owners did not differ from non-pet owners in walking for exercise or self-reported health. The positive relationship between dog ownership and health outcomes was not influenced by sex, race, or geographical living area. These findings are consistent with prior literature demonstrating a positive relationship between dog ownership and increased physical activity and better health [12,13,14,26]. It is possible that these effects are self-selecting, as people who are healthier may be more inclined and able to keep a dog as a pet. In contrast, cat ownership may be less physically demanding. 

Non-pet owners did not differ from dog or cat owners in levels of depression symptoms in the present study, which was inconsistent with study hypotheses. The current sample was taken from a community rather than a clinical population and had relatively low levels of self-reported depressive symptomatology, which may have contributed to a lack of findings between pet ownership and depression; however, other studies have also failed to find this relationship [8]. Additionally, there are many challenges that older adult pet owners face, particularly if there is a decline in their physical or cognitive health [27]. For example, cleaning up dog or cat litter may represent a barrier to pet ownership in those with poorer physical health [28]. Other challenges may include worry over making arrangements to care for a pet if a hospitalization or move to a residential care facility becomes necessary [29]. In addition, caring for a pet with behavioral problems may have a negative impact on the emotional health of the owner [30]. These findings imply that mood is a complex psychological construct and, while some older adults may benefit from pet ownership, adopting a pet is not without its challenges and should not be viewed as a simplistic solution for mental health issues.

The current study’s findings should be interpreted with respect to several limitations. The study was comprised of only White and Black participants and most participants had reported minimal symptoms of depression. Further, depression diagnoses were not assessed. Findings will thus benefit from replication in more racially heterogenous and clinically severe samples. Additionally, attachment to pets was not measured in this dataset, and future research may consider this important aspect of the human–animal relationship when examining the impact of pet ownership on mental health. Lastly, the study’s correlational observation design precludes drawing inferences regarding causality.

## 5. Conclusions

The present study is the first to investigate the relationship between pet ownership and mental and physical health benefits among older adults utilizing a sample balanced for sex, race, and rural or urban geographical location. Pet owners who owned dogs reported better overall health and more physical activity over the past year as compared to non-pet owners, while cat owners did not differ from non-pet owners on measures of health. The positive association between dog ownership and health was similar among males and females, Whites and Blacks, and those living in rural and urban areas.

## Figures and Tables

**Table 1 ijerph-19-05655-t001:** Participant Characteristics.

	Dog Owners (N = 116)	Cat Owners (N = 60)	Non-Pet Owners (N = 360)
Age at Recruitment	72.17 (4.92)	73.53 (6.04)	73.67 (5.74)
Sex	47.4% Female	55.0% Female	56.9% Female
Race	42.2% Black	33.3% Black	52.5% Black
Geographical Location	53.4% Rural	53.3% Rural	44.4% Rural
GDS-15 *	1.54 (1.72)	1.73 (1.68)	1.75 (1.70)
Self-Reported Health *	3.20 (0.58)	3.40 (0.69)	3.34 (0.64)
Self-Reported Health **	3.25 (0.52)	3.39 (0.53)	3.38 (0.64)
Walking for Exercise (within previous year) *	17.20%	18.30%	10.00%

Notes. * 84-month data collection; ** 90-month data collection; GDS-15 = 15-item Geriatric Depression Scale (higher scores = higher depressive symptomatology; range from 0 to 15), Self-Reported Health range from 1 (excellent) to 5 (poor).

**Table 2 ijerph-19-05655-t002:** Depression and Self-Rated Health by Sex, Race, and Geographical Location.

	Males	Females	Whites	Blacks	Urban	Rural
GDS-15 *	1.53 (1.66)	1.85 (1.72)	1.36 (1.41)	2.07 (1.91)	1.53 (1.66)	1.90 (1.73)
Self-Reported Health *	3.22 (0.59)	3.39 (0.66)	3.28 (0.63)	3.36 (0.63)	3.29 (0.65)	3.35 (0.62)

Notes. * 84-month data collection; GDS-15 = 15-item Geriatric Depression Scale.

## Data Availability

Restrictions apply to the availability of these data. Data were obtained from the Integrative Center for Aging Research and are available from https://www.uab.edu/icar/research/data-sets (accessed on 18 March 2022) with the permission of UAB Study of Aging investigators.

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
