# Peer review of "Pet Ownership and Mental and Physical Health in Older White and Black Males and Females"

_ijerph, 2022, doi:10.3390/ijerph19095655_

Round 1
Reviewer 1 Report
The Diagnosis of depression could have been substained not only by GDS but also with a diagnosis based on DSM V
The study is very general and it is not clear how the pet ownership could influence the simptoms of depressionn
the difference between the owner of dogs ad cats could be better explained
Also the importance of animal human relationship could have been investigated, particularly in relation to dogs owners. The paper can be improved explaining the importance of this relationship to mitigate depressive symptoms and not only stressing the better motor activity related to own a dog
Author Response
Reviewer 1:
Comment: The Diagnosis of depression could have been sustained not only by GDS but also with a diagnosis based on DSM V
Response: The reviewer raises an excellent point that the GDS is a screener and is not sufficient to make a diagnosis of depression based on DSM-5 criteria. Unfortunately, information regarding formal mental health diagnoses was not collected for the current dataset. We attempt to speak of depression “symptomatology” or “symptoms” throughout the manuscript and have added the following sentence to the methods section to clarify this: “Of note, the GDS is a screening measure, and, while it measures depressive symptomatology, significant scores (i.e., greater than 5) are not equivalent to a diagnosis of major depressive disorder according to DSM-5 criteria.”
We have also added a new sentence to the limitations section of the discussion that states:
“Further, depression diagnoses were not assessed.”
Comment: The study is very general and it is not clear how the pet ownership could influence the symptoms of depression
Response: Unfortunately, the only measure of depressive symptomatology collected in this dataset was the GDS and the sample had overall low rates of depression symptomatology with most participants endorsing few to no GDS symptoms. We are therefore unable to comment on how pet ownership influences specific symptoms of depression.
We noted in the introduction that pet ownership may help to alleviate loneliness and thus potentially feelings of depression. However, as noted in the discussion, this study did not find that pet ownership influenced depression symptomatology as measured by the GDS, which is consistent with other research regarding mental health and pet ownership in older adults (i.e., Gee et al., 2017).
Comment: The difference between the owner of dogs and cats could be better explained
Response: To clarify that there were no significant differences in terms of sex or rural/urban location, the following sentences have been added to the results section to clarify this:
“Dog and cat owners did not significantly differ in terms of geographic location. Older adults living in urban areas were not more likely to own a dog or a cat as compared to older adults in rural areas and vice versa. Similarly, sex did not influence type of pet owned, as males and females did not significantly differ from one another on their likelihood of owning a cat or a dog. “
Comment: Also the importance of animal human relationship could have been investigated, particularly in relation to dog owners. The paper can be improved by explaining the importance of this relationship to mitigate depressive symptoms and not only stressing the better motor activity related to own a dog
Response: We thank the reviewer for raising this point. The literature regarding the relationship between pet ownership and mental health is mixed (i.e., Gee et al., 2017 versus Krause-Parello et al., 2019), and there were no significant relationships found between pet ownership and depressive symptomatology in our study, which was inconsistent with study hypotheses. This was the case for both owners of cats and dogs. This has been emphasized in the discussion with the modified sentence:
“Non-pet owners did not differ from dog or cat owners in levels of depression symptoms in the present study, which was inconsistent with study hypotheses.”
As noted in the discussion, we believe this may be due to the non-clinical nature of the sample, and we have added a statement noting that participants were drawn from a community population. Unfortunately, attachment to pets was not measured in the UAB Study of Aging, so we are unable to comment on how this specific aspect of the human-animal relationship may influence depressive symptomatology. This has been added as a limitation in the discussion which now states:
“Additionally, attachment to pets was not measured in this dataset, and future research may consider this aspect of the human-animal relationship when examining the impact of pet ownership on mental health.”

Reviewer 2 Report
Thank you for the opportunity to review the manuscript.
In my opinion, this is an interesting study, but in a sense, the results obtained from it are predictable. The main findings are no different from those that can be expected. This article does not represent a substantial progress in knowledge, because the analysis is too simple, and it does not provide valuable data or draw valuable conclusions, which can be used to help improve the mental health of the elderly.
Author Response
Reviewer 2:
Comment: Thank you for the opportunity to review the manuscript. In my opinion, this is an interesting study, but in a sense, the results obtained from it are predictable. The findings are no different from those that can be expected. This article does not represent a substantial progress in knowledge, because the analysis is too simple, and it does not provide valuable data or draw valuable conclusions, which can be used to help improve the mental health of the elderly.
Response: We thank the reviewer for their comments and concerns. We want to emphasize however that the literature on pet ownership and physical and mental health outcomes is mixed (i.e., Gee et al., 2017; Krause-Parello et al., 2019), and positive outcomes are not consistently documented. Consequently, we believe that results from a study investigating the relation between pet ownership and mental and physical health are not predictable given the mixed nature of the literature. Many older adults own pets and many older adults have mental and physical health problems. Efforts to elucidate the relation between pet ownership and health in older adults and help address the mixed literature is, in our opinion, a worthy endeavor and helps to progress knowledge in a field where findings remain mixed. With respect to progression of knowledge and the analyses employed, when a relation in the literature is mixed, there are two common avenues that researchers take to investigate the potential contributors of the mixed relation. The two avenues are to identify and investigate potential mediators, and identify and investigate potential moderators. Our study incorporated use of moderation analyses as well as longitudinal analyses to investigate the relation between pet ownership and health. Though by no means were our analyses advanced on say the level of machine learning or SEM modeling, we believe that our moderation and longitudinal analyses were sufficient in addressing the questions proposed in the study and the findings of which helped to contribute to resolving the mixed literature through use of a unique sample with many strengths, i.e., large and balanced by race, sex, and geography. In particular, Black and rural participants are frequently excluded or underrepresented in research, and we believe that their inclusion in this study makes a substantial contribution to the existing literature. Lastly, we have made changes to the manuscript based on comments and suggestions provided by the other reviewers, and we hope that these changes help strengthen the manuscript and alleviate some of your concerns regarding the quality and importance of this manuscript.

Reviewer 3 Report
I enjoyed reading this manuscript and hope the authors find my comments helpful.
Abstract:
Line 21: I was a bit confused by the statement, “Findings were not moderated by sex, race, or geographical location”, mostly by use of the term “moderated”. The statement in the Discussion (lines 173-174), “The positive relationship between dog ownership and health outcomes was not influenced by sex, race, or geographical living area” was clearer.
Introduction:
The Introduction did a nice job of setting up the study.
Line 63: I would insert “8-year long” before “University of Alabama (UAB) Study of Aging” to provide context for your choice of using 84- and 90-month time points. I think it is okay to leave this information on duration of the study in line 78 of Material and Methods.
Lines 72-74: Predictions are included for symptoms of depression and self-reported health but not for physical activity; I suggest that you also include how you expected pet ownership to influence physical activity.
Material and Methods:
Lines 83-84: The paper is fairly short, so there is space for additional information on participant recruitment; readers should not be referred to the Baker et al. (2003) paper for information important to understanding and evaluating your study.
Lines 101-102: Does “walked for exercise” include walking a pet dog?
Results:
Tables should be “stand-alone”, meaning that a reader should be able to understand them without reading the text of the paper. In Table 1, for example, I would indicate in the Notes that for “GDS-15”, higher scores indicate more symptoms of depression and for “Self-Report Health” scores ranged from 1 (Excellent) to 5 (Poor).
Minor point: most journals require leading zeroes (e.g., 0.05 not .05)
Lines 132-135: Rather than summarizing the directions of results after presenting all the statistics, I suggest placing direction information with the specific statistics. For example, “At 84-months, data indicated a sex difference for self-rated health, with females, on average, having higher ratings (ANOVA F(1, 531) = 9.53, p < 0.01). There was also a sex difference in depression symptomatology, with females reporting more symptoms of depression (ANOVA F(1, 507) = 129 4.67, p = 0.03).” And do the same for the findings on race and geographical location.
Lines 156-157: Does “time” mean duration of walk? If so, I would explicitly state this to clarify for readers.
Discussion:
I would like to see a better integration of your findings with those from previous research. For example, in the paragraph running from line 168 to line 179, there are 8 lines of your findings until there is some comparison to other studies.
Lines 184-187: There are now some papers on how having a pet with behavioral problems impacts owners (see, for example, Buller and Ballantyne. 2020. Journal of Veterinary Behavior 37: 41-47). Although this is not something limited to older pet owners, you might want to include it in your list of challenges of pet ownership.
Finally, the list of references has spacing issues.
Author Response
Reviewer 3:
Comment: Line 21: I was a bit confused by the statement, “Findings were not moderated by sex, race, or geographical location”, mostly by use of the term moderated”. The statement in the Discussion (lines 173-174), “The positive relationship between dog ownership and health outcomes was not influenced by sex, race, or geographical living area” was clearer.
Response: The word “modified” has been replaced with “influenced” in the abstract to improve clarity.
Comment: Line 63: I would insert “8-year long” before “University of Alabama (UAB) Study of Aging” to provide context for your choice of using 84- and 90-month time points. I think it is okay to leave this information on duration of the study in line 78 of Material and Methods.
Response: The phrase “eight-year-long” has been included to provide context for the data collection timepoints.
Comment: Lines 72-74: Predictions are included for symptoms of depression and self-reported health but not for physical activity; I suggest that you also include how you expected pet ownership to influence physical activity.
Response: Thank you for raising this point. We have added a statement to our hypotheses clarifying that we predicted greater physical activity in older adults who were pet owners compared to those who were not. The last sentence of the introduction now states:
“Older adults who were pet owners are predicted to have lower symptoms of depression as well as higher self-reported health and greater levels of physical activity as compared to older adults who do not own any pets.”
Comment: Lines 83-84: The paper is fairly short, so there is space for additional information on participant recruitment; readers should not be referred to the Baker et al. (2003) paper for information important to understanding and evaluating your study.
Response: Thank you for this suggestion. The sentence referring readers to Baker et al. has been removed, and the recruitment and data collection process has been described in greater detail.
The additional details added to the Materials and Methods section now state:
“Participants were Medicare beneficiaries randomly recruited by letter from two urban and three rural counties located within central Alabama. All participants were community-dwelling, as residents of skilled nursing facilities were excluded from this study. In addition, individuals who were not able to independently schedule study appointments were excluded. Following obtainment of informed consent, an initial interview that lasted for approximately two hours was conducted by trained personnel within the participant’s home to collect initial baseline data. Following the initial interview, follow-up telephone interviews were conducted every six months for eight years”
Comment: Lines 101-102: Does “walked for exercise” include walking a pet dog?
Response: We have attempted to clarify this in the methods section through the addition of the following sentence:
“While walking for exercise was assessed as a separate variable from walking with a pet dog, it is likely that the walking for exercise variable also captures participants who exercised with their pets.”
Comment: Tables should be “stand-alone”, meaning that a reader should be able to understand them without reading the text of the paper. In Table 1, for example, I would indicate in the Notes that for “GDS-15”, higher scores indicate more symptoms of depression and for “”Self-Reported Health” scores ranged from 1 (Excellent) to 5 (Poor).
Response: Information regarding the ranges for GDS score and self-reported health has been added to Table 1.
Comment: Minor point: most journals require leading zeroes (e.g., 0.05 not .05).
Response: The manuscript and tables have been edited so that leading zeroes are present for all reported decimals.
Comment: Lines 132-135: Rather than summarizing the directions of results after presenting all the statistics, I suggest placing direction information with the specific statistics. For example, “At 84-months, data indicated a sex difference for self-rated health, with females, on average, having higher ratings (ANOVA F(1, 531) = 9.53, p < .01). There was also a sex difference in depression symptomatology, with females reporting more symptoms of depression (ANOVA F(1, 507) = 4.67, p = .03).” And do the same for findings on race and geographical location.
Response: Thank you for this suggestion. We have included the direction of results with statistical information, which hopefully provides additional clarity and makes this paragraph more efficient.
The paragraph now states:
“At 84-months, data indicated a sex difference in self-rated health, with females reporting better subjective health (ANOVA F(1, 531) = 9.53, ηp2 = 0.018, p < 0.01). There was a similar effect of sex on depressive symptomatology, with females reporting higher GDS scores (ANOVA F(1, 507) = 4.67, ηp2 = 0.009, p = 0.03). Depression symptomatology also differed between Whites vs. Blacks, as Black participants reported higher GDS scores (ANOVA F(1, 507) = 23.14, ηp2 = 0.044, p < 0.01). Similarly, individuals living in rural locations also had significantly higher GDS scores than their urban counterparts (ANOVA F(1, 507) = 17.28, ηp2 = 0.012, p = 0.01).
Comment: Lines 156-157: Does “time” mean duration of walk? If so, I would explicitly state this to clarify for readers.
Response: We apologize for the lack of clarity. In this case, “time” refers to the difference between the 84- and 90-month time periods. We have rephrased this sentence so it is more explicitly stated and the revised sentence now states:
“Repeated measures ANOVA predicting self-reported health across 84-months and 90-months found a significant main effect for dog ownership, i.e., owned a dog vs did not own a dog, (F(1, 451) = 6.37, ηp2 = 0.014, p = 0.01) and time, i.e., change in self-reported health from 84 months to 90 months (F(1, 451) = 6.04, ηp2 = 0.013, p = 0.01)”
Comment: I would like to see a better integration of your findings with those from previous research. For example, in the paragraph running from line 168 to line 179, there are 8 lines of your findings until there is some comparison to other studies.
Response: We have added additional integration of our findings with that from previous literature.
Our findings that older women have higher self-reported health and higher symptoms of depression is consistent with the literature, so we have added additional information to the beginning of the discussion noting this. Similarly, our results regarding depression in rural participants are also expected given past findings, and this is also noted. Further discussion is provided regarding depression in older Black adults.
The added discussion of these findings to the discussion section states:
“Older adult females reported on average better overall health but worse depression symptoms as compared to older males, which is consistent with prior research regarding self-reported health [15] and depression [16]. Black older adults and older adults living in rural locations reported on average worse depression symptoms as compared to Whites and individuals living in urban locations, respectively. This is consistent with prior research, which suggests that older adults living in rural locations are more likely to report symptoms of depression [17]. The literature is more mixed regarding depression in older Black adults, as Whites are more likely to be diagnosed with depression; however, it is likely that formal diagnoses are influenced by social determinants of health, and depression may simply be less recognized in minority populations [18]. Given that the current study used the GDS to assess symptoms of depression rather than a formal diagnosis of a mood disorder, symptomatology may be better captured by this measure and may explain higher rates of self-reported symptoms in Black participants.”
Comment: Lines 184-187: There are now some papers on how having a pet with behavioral problems impacts owners (see, for example, Buller and Ballantyne. 2020. Journal of Veterinary Beahvior 37: 41-47). Although this is not something limited to older pet owners, you might want to include it in your list of challenges of pet ownership.
Response: Thank you for providing this reference. This has been added to the challenges of pet ownership, as well as additional recent research regarding how cleaning up dog and cat litter may be a deterrent for older adults with poor physical health (Janevic et al., 2020).
The added discussion of these challenges and deterrents to the discussion section states:
“Additionally, there are many challenges that older adult pet owners face, particularly if there is a decline in their physical or cognitive health [21]. For example, cleaning up dog or cat litter may represent a barrier to pet ownership in those with poorer physical health [22]. Other challenges may include worry over making arrangements to care for a pet if a hospitalization or move to a residential care facility becomes necessary [23]. In addition, caring for a pet with behavioral problems may have a negative impact on the emotional health of the owner [24].”
Comment: Finally, the list of references has spacing issues.
Response: Thank you for noting this. Spacing issues have been addressed.

Reviewer 4 Report
This study presents the results of a survey the associations between pet ownership and physical/mental health in older people. Its true novelty is that it compares white and Black people; non-whites are under-represented in research of this kind, so the findings are a useful contribution to the field.
This is a nice, tight report, and I enjoyed reading it. I just have a few suggestions.
Throughout - the authors use two terms interchangeably: African American and black. I suggest that they pick one and stick with it. Should it be African American, black, or Black? A quick Google search left me more confused about the topic, but perhaps the authors know the general state of play on this currently.
Abstract - suggest noting the differences between the races in pet ownership, as this is a major finding which is not reported in the abstract.
L34 - mention of the pandemic seems out of place here because the data were collected long before COVID hit. I don't think it's necessary to use the pandemic as an excuse to do research in this area. Depression and loneliness were problems for older people long before the pandemic hit, and they will continue to be long after it's over.
Otherwise, good intro.
Table 1 - suggest adding the N's next to Dog Owners, Cat Owners, and Non-Pet Owners
Table 1 - suggest adding '...within previous year' after 'Walking for exercise'
Results - ANOVA effect sizes would be useful here, because there appear to be several sig outcomes that may not actually be particularly meaningful once effect sizes are noted (e.g., dog owner health vs non-owners was M = 3.2 vs M = 3.34).
L153 - 'non-sig trend' at p = 0.06, but at L149 there was a p = 0.06 that was just non-sig. Suggest removing the 'trend' in L153 - it's just not significant.
L170 - I'd like to see some discussion of the animal welfare implications about the finding that only 8% of participants reported walking their dog in the previous two weeks. This is a serious problem if only a small percentage of dogs are getting walked by their owners in this population. How can we help resolve this problem in practice?
Author Response
Reviewer 4:
Comment: Throughout – the authors use two terms interchangeably: African American and black. I suggest that they pick one and stick with it. Should it be African American, black or Black? A quick Google search left me more confused about the topic, but perhaps the authors know the general state of play on this currently.
Response: We thank the reviewer for this excellent and important suggestion. Based on current trends within the literature, we have decided to use the terms “Black” and “White” throughout when discussing participant race. The manuscript has been edited throughout to reflect this change, which also includes a minor change to the manuscript title.
Comment: Abstract – suggest noting the differences between the races in pet ownership, as this is a major finding which is not reported in the abstract.
Response: The abstract has now been updated to include this finding which now states:
“White participants were more likely than Black participants to report ownership of a pet.”
Comment: L34 - mention of the pandemic seems out of place here because the data were collected long before COVID hit. I don’t think its necessary to use the pandemic as an excuse to do research in this area. Depression and loneliness were problems for older people long before the pandemic hit, and they will continue to be long after it’s over.
Response: The reviewer certainly raises an excellent point that depression and loneliness were and will continue to be significant issues faced by older adults regardless of the COVID-19 pandemic. However, we thought it was important to include this sentence in the introduction given the rate at which pet ownership has increased due to the pandemic and a major finding of our study was that pet ownership did not influence depression. Given this, we hope that a takeaway from this study is that adopting a pet is not a simplistic solution to ameliorating symptoms of depression. These findings may have important implications for animal welfare during a time when high rates of pet adoption may be resulting in pets being adopted into inappropriate settings or homes.
Comment: Table 1 - suggest adding the N’s next to Dog Owners, Cat Owners, and Non-Pet Owners
Response: Ns have been added to Table 1 for each group.
Comment: Table 1 - suggest adding “…within previous year” after “Walking for exercise”
Response: This has been added to Table 1 to provide additional clarity.
Comment: ANOVA effect sizes would be useful here
Response: ANOVA effect sizes, i.e., partial eta squared (ηp2) have been added to the results section.
Comment: L153 – “non-sig trend” at p = 0.06, but at L149 there was a p = .06 that was just non-sig. Suggest removing the “trend” in L153 – it’s just not significant.
Response: This line has been edited to reflect that the finding was non-significant and the word “trend” has been deleted.
The sentence now states:
“However, dog ownership did not predict better self-reported health six months later at the 90-month follow-up, (ANOVA F(1, 452) = 3.5331, ηp2 = 0.009, p = 0.06).
Comment: L170 - I’d like to see some discussion of the animal welfare implications about the finding that only 8% of participants reported walking their dog in the previous two weeks. This is a serious problem if only a small percentage of dogs are getting walked by their owners in this population. How can we help resolve this problem in practice?
Response: We thank the reviewer for raising this point about animal welfare, and it would indeed be a serious issue and surprising finding if the vast majority of dogs in this sample were not getting adequate exercise. However, we think this may be explained by access to yards or public outdoor spaces, which is not uncommon in central Alabama. We have incorporated this into the discussion section, as well as research supporting this hypothesis (Toohey & Rock, 2011).
The added section discussing the low rates of dog walking in the discussion section states:
“It is important to note that although dog owners tended to walk more for exercise, only a small minority of them, 7.8%, reported walking their dogs over a two-week period. This was a surprising finding, given that increased exercise in the company of a pet is often a reason for adoption. However, research has demonstrated that access to yards or public outdoor spaces allows many dog owners to provide adequate exercise for their pet without being required to take them for walks [19]. It is likely that this explains, at least in part, the low number of participants who reported walking with their dogs, as these spaces tend to be fairly common and accessible in central Alabama.”

Reviewer 5 Report
Accept with major revision
This paper contains information about pet ownership that is important. The authors need to greatly improve their paper by providing more references to support their introductory statements and a more complete description is needed to describe their sampling methods. The data set they used is over 20 years old and the authors need to provide an explanation to defend that the samples is still relevant today.
Line 38 - Provide two additional references that depression is associated with negative mental and physical health.
Line 39 - Add a reference for the statement for the 5% figure.
Line 41 - Add at least one more reference for pets improving loneliness and depression.
Line 75-84 - There needs to be a more complete description of how your sample was recruited. Provide an explanation on why a sample that is over 20 years old would be still relevant today.
Line 92 - Reference for the Geriatric Depression Scale.
Line 97 - References for self reported Health and Physical Activity scale. If it was created by the researchers 20 years ago, please explain it.
Line 168 - You need to discuss the effects that low income may have had on your findings.
Line 181 - You also need to discuss possible reasons for a low level of self reports on depression symptoms.
Author Response
Reviewer 5:
Comment: Line 38. Provide two additional references that depression is associated with negative mental and physical health.
Response: As requested, we have provided two additional references: Brandolim Becker et al., 2018 and Carriedo et al., 2020.
Comment: Line 39. Add a reference for the statement for the 5% figure.
Response: We apologize for the confusion. This statistic is taken from the Fiske et al. reference, and this has been clarified within the text.
The sentence now states:
“Depression is associated with a host of negative mental and physical health outcomes among older adults [4, 5, 6], with a prevalence rate of up to 5% [4].”
Comment: Line 41. Add at least one more reference for pets improving loneliness and depression.
Response: We have included the additional reference of Hui Gan et al., 2020. The sentence now states:
“Although some studies have found beneficial associations between pet ownership and depression among older adults, this literature remains mixed [8; 9].”
Comment: Line 75-84. There needs to be a more complete description of how your sample was recruited. Provide an explanation on why a sample that is over 20 years old would still be relevant today.
Response: Thank you for this suggestion. The sentence referring readers to Baker et al. has been removed, and the recruitment and data collection process has been described in greater detail.
The additional details added to the Materials and Methods section now state:
“Participants were 1000 Medicare beneficiaries randomly recruited by letter from two urban and three rural counties located within central Alabama. All participants were community-dwelling, as residents of skilled nursing facilities were excluded from this study. In addition, individuals who were not able to independently schedule study appointments were excluded. Following obtainment of informed consent, an initial interview that lasted for approximately two hours was conducted by trained personnel within the participant’s home to collect initial baseline data. Following the initial interview, follow-up telephone interviews were conducted every six months for eight years”
Regarding the age of this dataset, we appreciate the reviewer’s concerns. However, depression and loneliness continue to be and, unfortunately, will likely remain important issues faced by older adults, particularly given the current pandemic. This dataset is quite unique and has many strengths (i.e., large and balanced by race, sex, and geography). In particular, Black and rural participants are frequently excluded or underrepresented in research, and we believe that their inclusion in this study makes a substantial contribution to the existing literature. We have made note of this in the introduction section through addition of the following sentence:
“This is a particularly important area of investigation given that Black and rural-dwelling individuals are frequently excluded from research, and thus may be underrepresented in the literature [16].”
We have also added the following information regarding the continued importance of this dataset:
“While data collection for this study began approximately twenty years ago, issues of depression and loneliness continue to be significant issues faced by older adults [6], and this topic remains highly relevant in the current era of COVID-19. As noted, physical and mental health concerns may be exacerbated by social isolation due to the current pandemic [18].”
Finally, pet ownership has increased significantly due to the pandemic and a major finding of our study was that pet ownership did not influence depression. Given this, we hope that a takeaway from this study for people today struggling with the pandemic and its related issues such as depression and who may be considering adopting a pet that adopting a pet is not a simplistic solution to ameliorating symptoms of depression. These findings may have important implications for animal welfare during a time when high rates of pet adoption may be resulting in pets being adopted into inappropriate settings or homes.
Comment: Line 92. Reference for the Geriatric Depression Scale.
Response: We apologize for the confusion and have moved the reference from the end of this sentence to make clearer that it is the citation for the GDS. The sentence now states:
“Depression. The 15-item Geriatric Depression Scale [GDS-15; 19] was used to collect information regarding depression at the 84-month assessment timepoint.”
Comment: References for self reported Health and Physical activity scale. If it was created by the researchers 20 years ago, please explain it.
Response: Information has been included stating that these questions were created by the original creators of the UAB Study of Aging. Participants were asked one question about self-rated health and two questions about walking for exercise, both of which are described in this section when discussing these measures. We have reworded this section to clarify these points:
“Self-Reported Health and Physical Activity. These constructs were assessed using questions created by the original authors of the UAB Study of Aging [17]. Self-reported health data from the 84- and 90-month time point was used in the current study. Participants were asked to rate their general health via a single question on a 5-point Likert scale, with options ranging from 1 = “Excellent” to 5 = “Poor.” Lower scores represent higher levels of self-reported health. Information about physical activity was taken from the 84-month assessment time point. Participants were also asked one question regarding if they had walked for exercise within the past year. If they indicated yes, they were then questioned regarding the average amount of time they walked on each occasion, the number of months in a year they walked, and how many times they walked during the past two weeks.”
Comment: Line 168. You need to discuss the effects that low income may have had on your findings.
Response: The reviewer raises an excellent point, as it is possible that income discrepancies may explain differences in those who can afford to keep pets versus those cannot. Unfortunately, data regarding pet ownership was taken from the 84-month timepoint, i.e., seven years into the longitudinal study. Data relating to self-reported income was collected during the initial baseline interview; as such, we are reluctant to make assumptions about the stability of this data over a seven-year period. It is highly possible that income of participants may have changed significantly since the time the study began. Stated simply, we unfortunately do not have data on participants’ income at the 84 and 90 month assessment timepoints when our variables of interested used in this study were assessed. We have attempted to note this through the following sentence in the discussion:
“While it is possible that this is potentially explained by discrepancies in income, as pets may be a significant expense, data regarding self-reported income is unfortunately not available at the 84-month timepoint and remains an area for potential future study.”
Comment: Line 181 – You also need to discuss possible reasons for a low level of self reports on depression symptoms.
Response: Thank you for raising this point. As this is a community rather than clinical sample, we believe that the non-clinical nature of the sample is the primary reason for the relatively low scores on the GDS. We have expanded the discussion section to include further discussion of our findings that Black participants were more likely to report higher GDS scores than Whites. In addition, we have added the following sentence to the discussion section:
“The current sample was taken from a community rather than a clinical population and had relatively low levels of self-reported depressive symptomatology, which may have contributed to a lack of findings between pet ownership and depression; however, other studies have also failed to find this relationship [8].”

Reviewer 6 Report
The paper is focused on association between pet ownership and health outcome.
I have few important concerns regarding the paper.
- The data are over 20 years old. As the topic is associated with public health and some psychosocial aspects of functioning, many things could have changed during that time.
- Study design: the Authors provide some data regarding the pet ownership and depression. There is a body of literature regarding positive effects of animal-assisted activities on mental health. The Authors do not however explore this scientific area. Moreover the Authors do no provide any rationale why out of many other mental disorders (dementia, anxiety, substances abuse) they chose to analyze depression.
- Discussion: the Authors do not discuss the simplest explanation of the obtained results: maybe fit persons who like walking obtain a dog while less fit and less active seniors do not like to take the dog ownership resposibility.
Author Response
Reviewer 6:
Comment: The data are over 20 years old. As the topic is associated with public health and some psychosocial aspects of functioning, many things could have changed during that time.
Response: Regarding the age of this dataset, we appreciate the reviewer’s concerns. However, depression and loneliness continue to be and, unfortunately, will likely remain important issues faced by older adults, particularly given the current pandemic. This dataset is quite unique and has many strengths (i.e., large and balanced by race, sex, and geography). In particular, Black and rural participants are frequently excluded or underrepresented in research, and we believe that their inclusion in this study makes a substantial contribution to the existing literature. We have made note of this in the introduction section through addition of the following sentence:
“This is a particularly important area of investigation given that Black and rural-dwelling individuals are frequently excluded from research, and thus may be underrepresented in the literature [16].”
We have added the following information regarding the continued importance of this dataset:
“While data collection for this study began approximately twenty years ago, issues of depression and loneliness continue to be significant issues faced by older adults [6], and this topic remains highly relevant in the current era of COVID-19. As noted, physical and mental health concerns may be exacerbated by social isolation due to the current pandemic [18].”
Finally, pet ownership has increased significantly due to the pandemic and a major finding of our study was that pet ownership did not influence depression. Given this, we hope that a takeaway from this study for people today struggling with the pandemic and its related issues such as depression and who may be considering adopting a pet that adopting a pet is not a simplistic solution to ameliorating symptoms of depression. These findings may have important implications for animal welfare during a time when high rates of pet adoption may be resulting in pets being adopted into inappropriate settings or homes.
Comment: Study design: the Authors provide some data regarding the pet ownership and depression. There is a body of literature regarding positive effects of animal-assisted activities on mental health. The Authors do not however explore this scientific area. Moreover the Authors do no provide any rationale why out of many other mental disorders (dementia, anxiety, substance abuse) they chose to analyze depression.
Response: Thank you for raising this point, and we certainly acknowledge that animal-assisted activities and therapies may certainly have a beneficial impact on mental health. However, we would like to note the significant difference between an animal trained to provide assistance (i.e., animal assistance dogs) versus those kept as pets. As, to the best of our knowledge, all animals in the study are pets rather than trained therapy dogs or cats, we are thus limited in what we can comment regarding the impact of animal assisted therapy/service animals on symptoms of depression using data from our sample. The following has been added to the introduction to clarify this distinction:
“Of note, the distinction between animals kept as pets and those trained in animal-assistance must be noted. While animal-assisted therapy may have significant positive impacts on mental health in older adults [i.e.,10], the literature is again mixed on whether typical pets without specialized raining may provide a similar effect.”
Additionally, while we also appreciate the reviewer’s comment that there are multiple other mental health diagnoses, such as schizophrenia or anxiety that would be interesting to address, depression was the only mental health variable collected in this dataset. However, given the mixed literature surrounding the effects of pet ownership on symptoms of depression, we hope that addressing depression adds to the current research.
Comment: Discussion: the Authors do not discuss the simplest explanation of the obtained results: maybe fit persons who like walking obtain a dog while less fit and less active seniors do not like to take the dog ownership responsibility.
Response: Thank you for this point.
We have stated in the discussion:
“It is possible that these effects are self-selecting, as people who are healthier may be more inclined and able to keep a dog as a pet. In contrast, cat ownership may be less physically demanding.” Within the discussion section, we have also expanded on other barriers to pet ownership that may cause those in poorer health to choose not to take on the responsibility of caring for a dog or cat.
Additionally, we have also further discussed potential reasons why such a small portion of the current sample reported walking with their dogs for exercise over the prior two weeks. This added addition to the discussion section now states:
“It is important to note that although dog owners tended to walk more for exercise, only a small minority of them, 7.8%, reported walking their dogs over a two-week period. This was a surprising finding, given that increased exercise in the company of a pet is often a reason for adoption. However, research has demonstrated that access to yards or public outdoor spaces allows many dog owners to provide adequate exercise for their pet without being required to take them for walks [25]. It is likely that this explains, at least in part, the low number of participants who reported walking with their dogs, as these spaces tend to be fairly common and accessible in central Alabama.”

Round 2
Reviewer 2 Report
I have understood your argument and I have reconsidered the sense of my evaluation.
Reviewer 5 Report
Accept this paper.Reviewer 6 Report
The main limitation of the study is that data were gathered 20 years ago. However, of course, nothing can be done about it.